# Realizing Sustainable Value from ERP Systems Implementation

**Luay Anaya [1,*]**, **Leif Flak [2]** and **Ahmad Abushakra [1]**

1    King Talal School of Business Technology/Business Information Technology, Princess Sumaya University for Technology, Amman 11941, Jordan
2    Department of Information Systems, University of Agder, 4630 Kristiansand, Norway
*    Correspondence: l.anaya@psut.edu.jo

**Abstract:** This paper investigates enterprise resource planning (ERP) implementations to improve the ability to realize outstanding value from such systems. In particular, it aims to provide a deep understanding of realizing sustainable value from ERP systems and to examine the relevance of benefits management (BM) in this context. To do so, this research applied a qualitative case study approach to investigate the implementation of Tier-1 ERP systems in two firms. Key findings initially suggest five considerations to better understand the realization of benefits from ERP implementation. Consequently, this research outlines the key activities undertaken by the investigated organizations and aligns them with activities suggested by benefits management literature. In conclusion, this research conjectures that while benefits management is a good practice and a systematic approach to realizing benefits from information systems, it may be ineffective in addressing the benefits that emerge in practice, i.e., when integrating the ERP system with modern digital technologies. Therefore, this research advocates either revisiting the current BM techniques or improving the implementation of digital technologies, including ERP systems with BM concepts and principles by incorporating such BM concepts within the implementation process. This study responds to research calls for maximizing the returned value from the implemented ERP systems by providing insightful recommendations.

**Keywords:** sustainable benefits; benefits management (BM); business value; benefits realization; Enterprise Resource Planning (ERP) system; digital technologies

## 1. Introduction

Businesses are progressively adopting advanced technologies and making significant investments to gain strategic advantages from these technologies. Among the most adopted technologies are Enterprise Resource Planning (ERP) systems. Such systems are considered essential technologies to many organizations as these systems can provide businesses with substantial advantages and more strategic business value than those which can be obtained from small Information Systems (IS) [1–3]. In particular, enterprises continuously adopt ERP systems to operationally solve business problems by facilitating the flow and dissemination of information and the automation of business processes and strategically fostering innovation and enabling business growth and sustainability [1,4–6].

Furthermore, ERP systems are playing a significant role in leveraging best practices, operational excellence, and organizational advantages for those firms that implement them [1,3,6]. A recent report [7] showed that ERP systems can provide companies with supply chain visibility to allow different parties to view information across the supply chain ecosystem, especially when they face troubled situations and unexpected levels of production demand for some products, as occurred during the recent COVID-19 pandemic. Consequently, ERP systems have become one of the most sophisticated and widespread technologies implemented in different types of enterprises, and they demand considerable financial and human resources, attention, and commitment [1,7–10].

It has been proven in existing research that while there are many benefits that can be realized from ERP systems, there are also several cases where enterprises were not

satisfied with the gained benefits, especially the strategic ones [10–12]. Instead, many firms experienced considerable challenges when attempting to realize substantial benefits from the implemented systems after delivery, which are referred to as post-implementation challenges [1,7,12–16]. A survey conducted by the "Project Management Institute" (PMI) suggests that barely a third of enterprises prioritize developing a comprehensive value-delivering capability [17]. It is sensible to note that the real success of implementing an IS project is not only delivering it on time and on budget; it is also important for enterprises to obtain strategic real value for their investment in IT/IS projects. Effectuating this returned value is still a problematic issue that faces organizations [2,18–22]. For example, Staehr et al. [1] considered some of the reasons for the difficulty in garnering return value for the investment of ERP systems, and the lack of realizing some of the ERP benefits can be attributed to the implementation process of such systems. This is because implementing new ERP systems involves a wide-range of organizational changes and the reengineering of business processes; some of these changes are difficult to manage and implement.

Numerous studies confirm that the ERP systems literature is rich in different perspectives, and there has been considerable progress in understanding different aspects of ERP implementation. However, realizing business benefits from these systems at the post-implementation stage is still problematic and puzzles several businesses; this has motivated the call for further research to address this problem [1–3,10,11,15,23]. Likewise, Pekkola and Päivärinta [14] have observed a lack of research into realizing benefits from ERP systems. They claimed that many of the information systems' success models reported in [24] identified in varied levels the benefit of realizing attention, but theorizing about the phenomena is still on an abstract level; therefore, they called for research to address this problem too.

Constructively, a body of research has been developed to respond to the aforementioned problem and to maximize the returned value of the massive investments on different information systems, including ERP implementation. In this research stream, authors have recommended applying a recognized approach or technique in the project management field, namely, "benefits management" (BM) [15,22,23,25–28]. Such a suggested approach can help organizations continuously identify their expected benefits and develop a plan to actualize these benefits in a well-structured process. It also underlines the importance of such an approach in addressing the technical and organizational factors that may hinder the realization of benefits. For example, ref. [22] argued that the literature has extensively investigated the different success factors that deliver a technical or a technological system, but it has fallen short of addressing the complicated issues face organizations after the system has been delivered. Indeed, many systems' benefits were obtained when they became integrated with other digital technologies; the benefits are not exclusive to a particular technological component that is isolated from the rest of the technological infrastructure [4–6]. Obviously, the implemented information system is integrated with other systems and with modern technologies within a digital transformation initiative [19,29].

Conversely, some studies find the BM approach to be a challenging endeavor [2,20,29,30]. Ref. [30] found that the benefits gained as a result of implementing ERP systems, specifically in small and medium enterprises (SME), are obvious or "self-evident", so it does not require formal techniques for benefits realization. In the same regard, other research [20] investigated the application of benefits management for IT projects in financial firms in particular, and they showed many barriers to applying IT BM. Therefore, the mentioned scholars have called for further research into applying IT BM.

Existing research has revealed that many enterprises implement ERP systems to help them cultivate strategic benefits and leverage business growth and sustainability over time. This study builds on a stream of research e.g., [4–7,11,18] that assumes that in order to achieve such advantages, enterprises should continuously work to exploit the technological advances and develop possible integrations between the implemented ERP systems and the modern technologies that emerge in practice after implementation. Most likely, these technological exploitations are challenging to perceive or identify before implementation,

as benefits management literature suggests. Therefore, this research aims to examine the relevance of the benefits management approach that suggests identifying expected benefits from ERP implementation, especially for organizations interested in realizing strategic benefits that span a long period of time. Such arguments, as well as previous research findings, provoke further investigation to deeply understand the main concerns and considerations related to realizing benefits from ERP systems. In this regard, this paper aims to investigate the following two research questions:

RQ1: "What are the central considerations concerning realizing the benefits of ERP systems?"

RQ2: "How does the benefits management approach differ from the existing practices undertaken by organizations to realize the benefits of ERP systems?"

## 2. Benefits Management

The terms benefits management (BM), benefits realization, and benefits realization management (BRM) are typically conflated and used interchangeably in the literature. While this is potentially problematic and should be addressed by the community, it is not the objective of this study. Our focus is on managing benefits and related approaches. We, therefore, use the term benefits management consistently throughout the paper.

BM has increasingly attracted the attention of the academic community as well as practitioners because it can offer a structured approach to improving the obtained value after carrying out projects or implementing technological systems [31,32]. To realize business benefits after implementing a technological system, BM suggests developing a plan to this end. Such a plan identifies the expected benefits and suggests critical measures to realize the expected benefits. It entails collective work toward achieving the benefits realization plan and emphasizes the critical role of management during the process. In this regard, scholars such as [15,22,27] have studied realizing benefits from IT/IS projects, and they sensibly argue that adopting or owning a digital technological product has no inherent value in itself and may not automatically generate business value. Therefore, the former scholars recommend that to realize the potential value of implementing an IT/IS, businesses can develop benefits management processes and continually work toward achieving the desired benefits. In essence, BM, as a body of research, as suggested by [33], seeks to shift the focus from the implementation of IS projects to the exploitation of these systems to create real value for investments in IS projects.

There are different benefits management models and frameworks. One of the most common models, called the "Cranfield Model", suggested in [27], is widely embraced in IT/IS research and projects. It can be considered that such a model, demonstrated in [27], is a most influential and an effective model for benefits realization [19,22,29]. Figure 1 shows the different stages for the BM process which start with the active engagement and involvement of both senior business managers and operational people to construct a BM plan that has details, such as the benefits' sources and their relation to the adoption motives, action responsibilities, required business changes, and timelines for achievements. These sub-processes are called "Identifying and structuring the benefits" and "Planning benefits realization". Subsequently, at the third stage, the developed plan is put into action to be executed; the results are reviewed and evaluated at the fourth stage; and lastly all stakeholders engage in a sub-process, called "Establishing potential for further benefits", within an ongoing process [27].

| Stage | Activities |
|---|---|
| 1 Identifying and structuring the benefits | • Analyse the drivers to determine the investment objectives.<br>• Identify the benefits that will result by achieving the objectives and how they will be measured.<br>• Establish ownership of the benefits.<br>• Identify the changes required and stakeholder implications.<br>• Produce first-cut business case. |
| 2 Planning benefits realization | • Finalize measurements of benefits and changes.<br>• Obtain agreement of all stakeholders to responsibilities and accountabilities.<br>• Produce benefits plan and investment case. |
| 3 Executing the benefits plan | • Manage the change programmes.<br>• Review progress against the benefits plan. |
| 4 Reviewing and evaluating the results | • Formally assess the benefits achieved or otherwise.<br>• Initiate action to gain outstanding benefits where feasible.<br>• Identify lessons for other projects. |
| 5 Establishing potential for further benefits | • Identify additional improvements through business changes and initiate action.<br>• Identify additional benefits from further IT investment. |

**Figure 1.** Benefits management stages and main activities [27].

Several principles illustrate the logic of BM, as shown in existing research [15,22,27,34]. One of the critical principles assumes that benefits realization is an actively continuing process. In other words, realizing benefits is considered a journey, not a destination, and business benefits are not outcomes automatically generated during the post-implementation of an IT/IS. The benefits may take time after the implementation, which entails collective efforts to work actively to identify these benefits and then manage the efforts to realize these benefits. Another principle considers some benefits to be outcomes that are not generated from a distinct or a stand-alone IT/IS, but stem from the complex interactions or integrations among a set of related digital technologies.

In essence, it is evident that ERP systems can help enterprises to achieve strategic goals and sustainable growth as outstanding benefits realized after the implementation [4–6]. At the same time, BM can be considered as a set of processes that incorporate what is necessitated of a business strategy to ensure that projects, programs, and portfolios create sustainable business value [34]. Accordingly, this paper aims to draw on BM as a theoritical stance to analyze and to relate the collected data based on its facets.

## 3. Research Methodology

To do this research, the authors adopted an interpretive case study, which assumes the reality of constructions highly related to people [35]. The undertaken interpretive philosophical approach enables the researchers to investigate the complicated and interconnected details related to realizing benefits from ERP systems by accessing the socially constructed knowledge from people who work in these systems. In fact, the case study strategy was applied in this research because it allowed us to explore practice and develop deep knowledge. Such approaches are particularly appropriate when the investigations seek to find answers to questions such as what, why, and how [35]. Qualitative data were collected to grasp people-dependent knowledge by understanding the social world from the viewpoints of social actors themselves, through reporting such data as detailed descriptions [36]. This research entails conducting a set of semi-structured interviews, as the primary data source is qualitative people-dependent knowledge [35]. Alongside the

interviews, triangulating the interview data with observations and documents analysis was also employed. Additionally, further discussions were conducted with external specialists who have experience in ERP systems and had participated in their implementations for different organizations. Such data sources allowed the researchers to make sense of the data collected and they provide a certain level of validity.

### 3.1. Data Collection and Analysis

The data were mainly collected from different people who participated in the implementation of Tier-I ERP systems (Oracle E-Business Suite) for two enterprises. As this research considers the reality as subjective, it was necessary to collect data from varied people working on different business functions, as most likely do not necessarily have the same perspectives on realizing the benefits of ERP systems. Such varied representation of different voices was vital to the investigation as it can help to avoid data bias [36]. Accordingly, and as is shown in Table 1, data have been collected from interviewees who have different business roles such as functional and technical consultants, project managers, head of departments, accountants, and system's users working at different business functions; different interviewees have different roles, backgrounds, experiences, and varied levels of involvement in the ERP system implementation. The duration of interviews was about 60 to 100 min for senior or managerial roles, and about 20 to 30 min for the systems' users.

**Table 1.** List of interviewees from the two cases.

| A-Case-1 | | B-Case-2 | |
|---|---|---|---|
| Int. Code | Interviewee Role/Position | Int. Code | Interviewee Role/Position |
| A1 | Financial director and internal project manager | B1 | Chief Financial Officer (CFO) and project sponsor |
| A2 | Head of reconciliation and accounts receivable | B2 | Head of accounting section and functional consultant |
| A3 | Head of fixed assets and inventory | B3 | Financial accountant |
| A4 | Accounts payable supervisor | B4 | Inventory and fixed assets accountant |
| A5 | Head of general accounting section | B5 | Technical consultant and application administrator |
| A6 | Payroll accountant and HR coordinator | B6 | Head of human resources section |
| A7 | Techno-Functional consultant | | |
| A8 | Finance coordinator | | |
| A9 | Technical team leader | | |
| A10 | ERP implementer | | |
| A11 | E-Business suite manager | | |

As noted earlier, this research aims to provide improved understanding of how to realize sustainable value from ERP systems and to examine the relevance of BM in this context. Consequently, the selection of cases is considered to support this research objective. In this regard, Hasan and colleagues [8] suggested that ERP implementation can be associated with different sustainability measures related to economic, environmental, and social dimensions. Hence, we investigated the implementation of ERP systems in two cases and monitored the impact of the implementation of the ERP systems for a decade to interpret some sustainability measures. The two cases have affected a number of sustainability measures that will be demonstrated later in the results section.

These cases are examined over a decade as the implementation of the ERP system started in 2007 for Case-1, and in 2009 for Case-2. The two firms showed consistent growth, as is shown in Table 2. Furthermore, these two companies had implemented ERP systems with a high level of satisfaction, and both companies sustained its use in the market and continued its development after the implementation. The two systems in these two cases were actively used, and the business operation is highly depended upon. While the ERP systems in the two cases helped their businesses to grow and to be sustained in the market, there was some context variation: Case-1 was a well-established firm; whereas Case-2 was a newly established firm when they implemented their ERP systems. Accordingly, it is assumed that the selected cases can provide a viable setting for answering the research questions of this study and provide insights relevant to achieving the study objective.

**Table 2.** Details of the two cases under investigation.

| Case Feature | Case-1 | Case-2 |
|---|---|---|
| Industry Type | Mobile Telecommunication | Mobile Telecommunication |
| The firm when it implemented the ERP implementation | Well-established | Newly established |
| Who was the project director | Business- Director of Finance Department | Business- Chief Financial Officer |
| Number of subscribers when the ERP was implemented | In 2007 about 1 million subscribers | In 2009, the business operation was just started |
| Number of subscribers in 2012 | 2.45 million subscribers | 600,000 subscribers |
| Number of subscribers in 2018 | 3 million subscribers | 1.25 million subscribers |
| ERP system | Oracle E-Business Suite | Oracle E-Business Suite |
| Legacy systems | Financial system, billing systems, other systems. | Basic systems and tools |

In addition to the interviewees from the two cases, three external experts provided further details related to the investigations as follows: ERP Consultant and Solution Architect (C1); Project Manager (C2); and Senior Technical Consultant (C3).

*3.2. Data Analysis*

The collected data were initially analyzed to highlight thought-provoking themes from an individual interview. The next stage was combining the dominant themes to articulate the participants' interesting issues. Specifically, for RQ2, the generated themes are aligned to relevant BM processes or activities that were each given a code, as shown in Table 3. When we found associations between the empirical data (themes) and the theoretical concept (BM code), the BM activity was then written in italics. For instance, we found the two cases developed RFP that demonstrated the main objectives of the ERP implementation. Such a theme has been aligned to "Analyze the drivers to determine the investment objectives" which was given a code (IB1) and written in italics. Similarly, both cases developed business cases, which were aligned with "produce business cases" (IB5), and which were also written in italics. Otherwise, BM activities that did not associate with existing practices are just documented under their relevant processes without being in italics. However, Figure 2 provides illustration about how the overall investigations are accomplished

**Table 3.** Summary of some actions undertaken by the cases which are aligned with BM facets.

| BM Processes and Activities | Examples about the Existing Practices Undertaken by Organizations |
|---|---|
| **Identifying and structuring benefits (IB)**<br>- *Analyze the drivers to determine the investment objectives * (IB1)*<br>- Identify the expected benefits (IB2);<br>- Establish ownership of benefits (IB3);<br>- Identify the changes required (IB4);<br>- *Produce business cases (IB5).* | • Each case developed a request for a proposal (RFP) that included the essential features and main requirements that should be embedded in the ERP system;<br>• Both cases developed business cases, and, as part of these cases, organizations identified some benefits before the implementation. |
| **Planning benefits realization (PB)**<br>- Finalize measurements of benefits (PB1);<br>- *Obtain agreement with stakeholders (PB2);*<br>- Produce benefits plan (PB3). | • Each case prepared an experienced team led by a competent project manager, which was a critical factor before the real implementation;<br>• Different stakeholders (project manager, implementation team, users) were discussing the different possibilities and advantages the ERP system could provide for the organization;<br>• Both cases chose the prescribed vendor methodology from Oracle as an implementation methodology that guided all efforts toward successfully implementing the ERP systems. |
| **Executing the benefits (EB)**<br>- *Manage change programs (EB1);*<br>- Review progress against the benefits plan (EB2). | • Both cases benefited from the total integration, from the best/good practices equipped with the ERP system to replace the existing ones, and from conducting business processes in ways different than those which they were used to (e.g., "end-to-end" business processes such as "procure-to-pay" and "order-to-cash" among others); in other words, cases managed to introduce set of business changes while the business was growing;<br>• The system configuration and customization were accomplished in such a way as to address the current and expected future needs of the organization. |
| **Reviewing and evaluating the results (RB)**<br>- Formally assess the benefits achieved (RB1);<br>- *Initiate actions to gain outstanding benefits (RB2);*<br>- *Identify lessons for other projects (RB3).* | • The ERP system helped a company (Case-1) analyze the revenue in different ways. The company found that it was possible to deal with different satellite towers as different revenue centres, so a new way to analyze the revenue emerged after the ERP implementation;<br>• Interviewees from Case-2 reported that they were conducting a meeting every week to discuss what the team accomplished, what work will be conducted, and how to address the requirements documented in the project file. |
| **Establish potential for further benefits (FB)**<br>- *Identify additional improvements (FB1)*<br>- *Identify additional benefits from further IT investment (FB2).* | • Each case formed a specialized team, including experts involved in the implementation early on, and this team suggested different ways to solve the emerging problems;<br>• The strong business relationships between the implementer companies and the cases have provided a solid base for the cases to grow, to be sustained, and to realize further benefits from the implemented ERP systems;<br>• In both cases, benefits accumulated from a portfolio of integrated systems, not just through realizing benefits from a single system. Case-1 acquired further benefits when it implemented CRM and BI systems into the post-implementation of the ERP system;<br>• The two cases kept the ERP system's configuration flexible, to be ready for further integrations or developments. |

*\* BM activities represented in italics are seen as in line with the existing practices undertaken by the cases.*

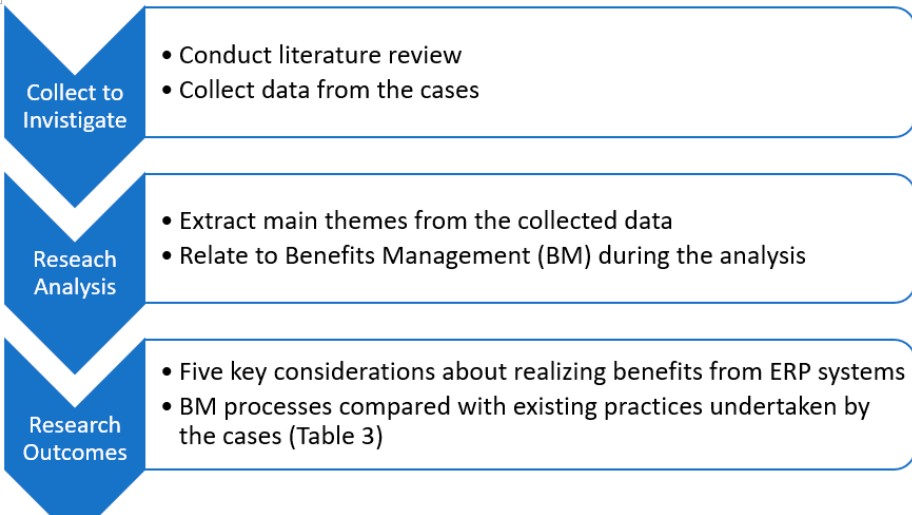

**Figure 2.** Research design.

## 4. Results

In this section, we aim to illustrate the key findings shown after conducting the research investigations. Initially, this study reported that the interviewees from the two cases were satisfied with the overall experience and the outcomes obtained as a result of implementing their ERP systems; they acknowledged gaining sustainable advantages or benefits without the formal adoption of BM approaches. Mainly, the ERP systems in the two cases helped the enterprises to sustain business growth, as shown in Table 2. In addition to that, ERP systems provided many benefits such as integrating business processes across different business functions, dealing with multiple users across different geographical places, providing information visibility, and allowing the system to provide best practices for many business processes, among other benefits. This section will firstly show the key considerations needed to understand realizing the benefits from ERP systems. It will then compare the actions undertaken by the organizations and the benefits management approach.

### 4.1. Achieved Sustainable Value

It has been reported in the literature that ERP implementation is related to a set of sustainability measures. In fact, ERP systems are likely to be instrumental as a backbone for registering corporate sustainability measures that can be extracted, e.g., for sustainability reporting. While this offers numerous opportunities for research, it was not the objective of this study. However, we will highlight examples about the sustainable value achieved in the cases and continue our investigations to address the efforts made during and after the implementation of ERP systems. In this paper, we refer to Hasan et al.'s [8] research, which provided examples of sustainability measures in economic, environmental, and social indicators that can be achieved by the implementation of ERP systems.

Initially, the two cases had successfully implemented ERP over more than a decade, the people working at the two firms were satisfied, and they are using the implemented ERP systems effectively. Such successful implementation is considered a sustainable business value as the firms are using the systems and they did not waste the efforts and investments made during the implementation. Furthermore, measures such as productivity, market share, and revenue are among the economic sustainable measures reported in a study [8]. The above-mentioned measures were found in the cases (see Section 3). In particular, Table 2 showed how the number of customers were growing over a long period of time. In terms of social measures, there are some indicators, such as career development, job opportunities, and number of employees that were developed over the last decade. For environmental measures, creating efficient and full automated business processes has tremendously reduced the consumption of paper. Moreover, providing self-service

transactions offered to staff, suppliers, and customers reduced physical attendance in the companies, which in turn reduced $CO_2$ emissions.

*4.2. Key Considerations about Realizing Benefits from ERP Systems*

Based on the empirical data, we identified several key considerations that represent the answer to the first research question (RQ1). These considerations are the following.

First, lack of awareness about formal benefits management techniques: This research shows that businesses do not employ formal techniques, such as BM, to realize business benefits from the implemented information systems. Clearly, the findings revealed that the investigated companies were not aware of BM techniques and the role they could play, as most of the interviewees expressed that they had not come across such techniques. For instance, a consultant (A7) from Case-1, who participated in implementing many ERP systems, reported, "I heard a lot about project management methods. However, this is the first time I hear about benefits management methods". Several interviewees have also expressed their lack of knowledge about benefits management methods (e.g., A9, A10, B5, and C3). Remarkably, the project leaders in both cases (A1, B1) acknowledged the importance of identifying some benefits before the implementation as part of developing business cases to obtain approvals for implementing ERP systems.

Second, difficulty in identifying benefits and quantitative measures early: The BM approach, as demonstrated in Section 2, suggests that businesses identify their expected benefits and outline quantitative measures or indicators to evaluate the performance level for these benefits. However, based on interviewees' responses, people face a difficulty in the early stages of determining the detailed benefits that are expected. It may not even be ready to define quantitative measures to monitor the performance level for these benefits. For example, one of the investigated organizations had just started its business operation when it implemented an ERP system (Case-2). The project sponsor (B1), who was the chief financial officer, said, "before the implementation, we were not having a full understanding about the benefits of the ERP systems, but we observed that many organizations, especially the successful ones have implemented ERP systems, so we thought that the system will help us too in our business development". Indeed, start-up companies, or some small and medium enterprises (SMEs), do not fully understand large-scale enterprise-wide systems such as ERP, their potential, and how they can benefit from the equipped capabilities. It is difficult for such start-up companies or SMEs to identify the expected benefits and define the performance indicators for those benefits. Such kinds of enterprises are interested in adopting the ERP system functionality and its embedded logic.

Furthermore, the study maintained that ERP systems differ from the development of information systems, which are developed for certain purposes. It is assumed that when organizations are developing new information systems, they may effectively adopt a benefits management approach to plan their needs, consider these needs through the system design and developments, and then monitor the system results. However, many interviewees in this research (e.g., A3, A6, B2, B3, and B4) considered the ERP system a product adopted to bring best practices to their organizations. Therefore, the BM is most probably challenging to be used formally in the ERP implementation because the benefits from ERP systems are seen as generated outcomes and not as something that could be clearly and completely planned in the early stages.

Practically, investigations found what was possibly a critical enabler for benefits realization from ERP system. It can be forming a specialized team who had started the implementation since early times and keep working together to provide support, developing well-crafted reports, and implementing further modules in addition to developing integrations with other systems or adding features. The project managers (A1, B1) in both cases emphasized the importance of formulating a qualified team with experienced skills and business and technical competencies. A project manager (A1) said it clearly: "[A] strong project manager who is competent and experienced is a key aspect in obtaining value from ERP systems". He continued, "The project manager and his team can highlight



the main benefits for the organization implementing an ERP system, in turn, the firm trusts the team's capabilities and empowers the project manager to help them exploit the ERP system's features and potential". An external project manager who served as an expert in this research (C2) has emphasized the vital role of the project manager in driving business value from the ERP system. Another consultant (B2) emphasized the critical role that having a well-developed and competent team from the beginning of the implementation can play. He stated: "Selecting the team members who have expertise in ERP implementation and allow the team to involve in the early stages of ERP implementation can develop the sense of ownership, and can drive organizations to the effective exploitation of the ERP system with minimal resistance".

Furthermore, as revealed by the interviewees, having a strong business relationship between the implementer companies and the cases has provided a solid base for the cases to realize further benefits from the implemented ERP systems. While Case-2 had developed a long-term support contract with the implementer company, Case-1 and the implementer company have a business relationship, as both are working under same business group.

Third, benefits emerged during practice: Many benefits obtained from ERP systems emerged in practice, after the implementation. For example, the financial director (A1) said that "the implementation of the ERP system has introduced to us new practices and ways to do our business operation differ than what we get used to". He continued, "look to the new end-to-end business processes such as procure-to-pay or order-to-cash that were impeded in the system. We did not fully understand such processes and the impact of these practices until later stages when we found the system brought total integration between the different modules". Similarly, the head of general accounting section (A5) stated that "we thought that the revenue analysis could be based only on the customer type. Still, after implementing the ERP system, we figured out that the system could provide great opportunities to analyze the revenue in different ways. For example, we realized that it was possible to deal with different satellite towers as different revenue centers", so the new way to analyze the revenue emerged after the ERP implementation. As pointed out above, the empirical investigations of this study revealed that people consider the detailed benefits to be the desired outcomes of the effective use and effective exploitation of the system's possibilities, but they are difficult identify completely in the early stages. Hence, this study advocates that organizations can establish a set of broad benefits or define a strategic direction based on their wide-ranging needs and expectations, and not necessarily predefine detailed benefits. Likewise, a techno-functional consultant (A7) who was coordinating with the marketing department to suggest certain promotion schema informed us that "analyzing the existing data to extract insights can help us to design effective promotions".

Furthermore, the interviewees (e.g., A10, A11, and B5) have also stressed the importance of adopting an implementation methodology that guides all efforts toward delivering the successful implementation of the ERP systems. Data collected from the different cases also show that every ERP system is implemented based on an agreed methodology. A solution architect (C1) who had participated in the implementation of ERP systems to many firms reported that "Different consulting companies choose different methodologies, some companies choose the prescribed vendor methodologies whether from SAP or Oracle or from other vendors, whereas other companies have their methodologies that draw a roadmap for the whole implementation stages".

Fourth, two management processes may implicate some complications: As the above findings exhibited, firms do not set formal BM methods or techniques in place either because they are unaware of such techniques or because it is challenging to apply them in some contexts. However, interviewees assumed some concerns about having two management processes: one for the ERP implementation life cycle and the other for BM. For example, as the e-business suit manager (A11) puts it, "If we adopt benefits management approach, who will be responsible for implementing that management process? And how to coordinate the tasks between the ERP implementation team, which some members are external consultants

from one side, and the team or staff that should work in benefits management approach from the other?". Therefore, different tasks and different responsibilities are called for by these different management processes.

Fifth, some business benefits do not come from distinct systems such as ERP systems: In some cases, organizations implement ERP systems as part of a digitalization process or a transformation endeavor. In this respect, the benefits are collected cumulatively and not from a distinct system. Indeed, ERP systems are implemented to complement other systems, so it may be difficult to realize the benefits from a single system (ERP). Data collected from the cases reported that the organizations wanted to implement ERP systems alongside other systems. The ERP project sponsor (B1) from Case-2 mentioned that "ERP system has been implemented because it is a fundamental system constitutes with other systems a technological infrastructure to help the company in its development and growth, and to give us the ability to plan to our business and respond to the customers' needs". He continued, "Our role is to implement the ERP system effectively and keep it flexible, and to be ready for further integrations or developments". In contrast, a project manager who had participated in the implementation of ERP to many organizations (C2) said that "The real value obtained from the implemented ERP system was after the implementation of ERP system, when the ERP system is complemented and integrated with other systems such as CRM and Business Intelligence (BI) systems to help them take benefit from the stored data". It seems that the ERP system has been considered as a pillar system that supports and complements other digital technologies, so the benefits are not effectuated from one distinct system [29,37].

*4.3. Actions Undertaken by Organizations Compared with Benefits Management*

To answer the second research question (RQ2), and after reporting some of the considerations and concerns that were experienced by the cases, this research continues to demonstrate the activities undertaken by the investigated organizations and how are they relevant to benefits management notions, which are shown above in Section 2, and particularly in Figure 1. Such a comparison is articulated in Table 3.

**5. Discussion**

Findings suggest a lack of awareness about the capacity of formal techniques to realize benefits, a matter clearly demonstrated in existing research [20,22,28,30–32]. In ERP systems, organizations expect that implemented systems will bring good, if not the best, practices to their businesses. Because of this merit, they wonder why they need to implement techniques for benefits realization. This research proposes that organizations can adopt any planning approach or methodology to work as a controlling tool to keep the project's focus. It can be the undertaken project management method or the ERP vendors' implementations methodology. In this regard, organizations may incorporate some activities or principles from the benefits management approach, presented in Section 2, within the adopted methodology without the formal adoption to BM techniques.

Several activities and practices undertaken by the organizations are aligned with the trajectories of benefits management. Mainly, the principle of BM that considers realizing benefits is an ongoing process, a matter that was found as a limitation in ERP literature. Existing ERP literature (e.g., [12,13,16]) found that the post-implementation stage is a complicated stage in implementing the ERP systems, and many organizations failed to obtain real value after delivering the ERP system. A study [16] found that the implementation team is disbanded when ERP implementation is finished, which usually obstructs enterprises from gaining the benefits of ERP systems.

This study's interviewees reported some concerns about having two management processes: one for the ERP implementation life cycle, suggested by the vendor, and the other for BM. Accordingly, having one inclusive management process for ERP implementation as a main management process, leveraged with good practices imported from BM, is proposed in this research to remove threats that may arise because of the application

of two different management processes. Interestingly, some implementation or project management methodologies have started to address the benefits realization efforts within their methodologies. For example, the "Project Management Institute" (PMI) published in 2019 a practical guide for benefits realization called "Benefits Realization Management (BRM) Practice Guide" [17].

However, this study strongly supports recent research [37] that found benefits accumulated from digital technologies such as data analytics, artificial intelligence, or smart-phones pose some challenges for managing benefits in that it is difficult to specify them prior to technology deployment. We similarly conclude, as later research [37] reported, that while BM is an established notion in the project management and technology literatures, it is not well-known as an organizational practice. Because of this, researchers [34,37] called for the BM approach for digital transformation programs to be revisited and adapted. Indeed, some enterprises implement ERP systems integrated with other cloud-based and mobile services to transform such firms. Accordingly, this research recommends improving the ERP implementation process by taking advantage of the different BM concepts without the formal adoption of the approach that stipulates defining detailed benefits in the early stages. The improved process may entail clear planning tasks, including defining a set of broad or strategic expectations, but it does so with less detail and without predefined measures for the system's expectations.

Taking all of this into account, this research supports previous studies (e.g., [1,3]) in that realizing the benefits of the ERP system is considered a challenging endeavor that will likely face organizations after they have implemented their ERP systems. Moreover, this study extends existing knowledge by providing rich insights about what organizations do to realize the substantial benefits of the ERP system. It also provides analysis for such practices and comparisons with what is suggested by BM literature, as illustrated in Table 3. It is conjectured here that identifying a list of benefits early, before implementing ERP systems, may present some complications for organizations that adopt such systems in search of a modern way to do their businesses in effective manner. In this regard, a recent study [37] recognized that identifying a "granular list of benefits" does not necessarily recognize the bigger picture of the organizations' long-term and strategic goals. When we compared activities undertaken by the investigated organizations, and the activities suggested by the benefits management literature, which were reported in Section 2 (and particularly in Figure 1), this research found some correspondences, which were reported in italics in Table 3. Accordingly, this study concludes that while businesses do not put in place formal approaches or methods to realize benefits from ERP systems, BM concepts and principles can be incorporated to improve the implementation of digital technologies. This study is in line with Breese's research [38], which found "benefits realization management is neither a panacea, nor a false dawn, but lies somewhere in between". Finally, this research advocates scholars' suggestion that "benefits management as a method to help realize benefits needs to be reinvigorated and re-imagined through a strategic lens" [37].

## 6. Conclusions

This research demonstrated a deeper level of understanding about realizing the sustainable benefits of ERP systems by showing a set of considerations. Realizing the benefits of ERP systems is not necessarily similar to the more general development of IS, which are developed for specific purposes. In IS development, organizations may effectively adopt BM techniques to plan their needs, consider them through the system design and developments, and then monitor the systems' results. On the other hand, the ERP system has been perceived as a product adopted to bring best practices to organizations. This study thus provides some empirical evidence to show the peculiarity of realizing the benefits of ERP systems.

Overall, this research argues that while benefits management is a good practice and a systematic approach to realizing benefits from information systems, it may be ineffective in addressing the benefits that emerge in practice, i.e., when integrating the ERP system with

modern digital technologies. Therefore, this research advocates either revisiting the current BM techniques or improving the implementation of digital technologies, including ERP systems with BM concepts and principles by incorporating such BM concepts within the implementation process. By doing so, this research provides rich insights and an improved understanding in addition to proposing some theoretical assumptions about realizing benefits from ERP systems as a theoretical contribution. Furthermore, this research offers great suggestions for practitioners working in ERP implementation. However, it may be interesting for future research to comprehensively investigate how organizations realize strategic benefits and undergo organizational changes when they implement ERP systems alongside other digital technologies within a digital transformation journey. This research has some limitations, as it relied on two cases with a limited number of interviewees. It would be interesting to pursue further research with a large number of cases working in different contexts to examine different contextual calibrations of best practices.

**Author Contributions:** Conceptualization, L.A. and L.F.; Methodology, L.A.; Investigation, L.A.; Writing–review & editing, L.A., L.F. and A.A. All authors have read and agreed to the published version of the manuscript.

**Funding:** This research received no external funding.

**Informed Consent Statement:** Not applicable.

**Data Availability Statement:** Not applicable.

**Conflicts of Interest:** The authors declare no conflict of interest.

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
