# Peer review of "Realizing Sustainable Value from ERP Systems Implementation"

_sustainability, doi:10.3390/su15075783_

Round 1

Reviewer 1 Report

I liked the research question and I believe that the conclusions are equally compelling. Overall, I believe this is an interesting topic and it has a sound structure. It does a good work at describing the main considerations concerning realizing benefits from ERP systems and how benefits management approach differs from existing practices undertaken by organizations to realize benefits from ERP systems. 

However, I believe the methodological part can be improved in order to gain further insights from the phenomena. Ideally, this could be done with a significant sample survey, but that might be outside the scope of the authors' approach. However, a second type of qualitative study can be employed in order to validate the insights gained from the current study. In its present form, I believe that the data qualifies more as an interesting case study, not as robust scientific study. Also, the results should better highlight the explanation of the phenomenon under study.

Also, the paper needs to highlight further its real contributions to the current literature. Please relate also with the aims and scope of the journal "Sustainability".

Good luck with your work.

Author Response

I would like to thank Reviewer1 for the insightful comments that really improved the paper. Please find bellow our responses to the comments.

Reviwer1:

I liked the research question and I believe that the conclusions are equally compelling. Overall, I believe this is an interesting topic and it has a sound structure. It does a good work at describing the main considerations concerning realizing benefits from ERP systems and how benefits management approach differs from existing practices undertaken by organizations to realize benefits from ERP systems. 

However, I believe the methodological part can be improved in order to gain further insights from the phenomena. Ideally, this could be done with a significant sample survey, but that might be outside the scope of the authors' approach. However, a second type of qualitative study can be employed in order to validate the insights gained from the current study. In its present form, I believe that the data qualifies more as an interesting case study, not as robust scientific study. Also, the results should better highlight the explanation of the phenomenon under study.

Response: We have added in red color some paragraphs to show a clear research aim and other paragraph regards choosing case study, and how case study strategy is chosen to meet the research objective. We have collected additional data to show the sustainable value achieved.

Also, the paper needs to highlight further its real contributions to the current literature. Please relate also with the aims and scope of the journal "Sustainability".

Response: in both introduction and findings sections We referred to the sustainability in a better way now

Many thanks for your support

Reviewer 2 Report

Dear authors,

First, thanks for the opportunity to read this research. Despite being interesting, I would recommend some adjustments that could improve its overall quality and readability.

Major improvements:

1. Despite being qualitative research, authors should consider including, for clarification purposes, the research and analysis model to allow readers to better follow the results and discussion; a figure would be highly useful in this context;

2. following this suggestion, I would also recommend authors split the "results and discussions" into two separate parts, improving the last one;

3. authors should also clarify what is the research gap and the paper's contribution to literature and its social and/or practical implications; it is not clearly stated in my humble opinion;

Minor improvements:

1. There are some redundancies that can be avoided, for instance: "The benefits management literature suggests that the BM as an effective approach to realize benefits can help organizations continuously identify their expected benefits and develop a plan to actualize these benefits in a well-structured process."

2. The same example also illustrates a different problem within the paper, related to the non-proper use of abbreviations.

3. I would suggest authors reproduce figure 2-1;

4. some excerpts might be revised and simplified (too long sentences), maybe splitting into two parts, improving their readability, for instance, the following: "Numerous studies confirm that while the ERP systems literature is rich in different perspectives, and while there has been considerable progress in understanding different aspects of ERP implementation, realizing business benefits from these systems at the post-implementation is problematic and puzzles several businesses, which has motivated the call for further research to address this problem [1-3,10,11,15,23]." 

Author Response

I would like to thank the Reviewer2 for the insightful comments that really improved the paper. Please find bellow our responses to the comments.

Reviwer2:

Dear authors,

First, thanks for the opportunity to read this research. Despite being interesting, I would recommend some adjustments that could improve its overall quality and readability.

Major improvements:

  1. Despite being qualitative research, authors should consider including, for clarification purposes, the research and analysis model to allow readers to better follow the results and discussion; a figure would be highly useful in this context;

Response: a new figure is added (Figure3-1) page5

  1. following this suggestion, I would also recommend authors split the "results and discussions" into two separate parts, improving the last one;

Response: Now Findings and discussion in separate sections

  1. authors should also clarify what is the research gap and the paper's contribution to literature and its social and/or practical implications; it is not clearly stated in my humble opinion;

Response: In the introduction section, a new paragraph to show the expected contribution/aim of the study with other parts added

Minor improvements:

  1. There are some redundancies that can be avoided, for instance: "The benefits management literature suggests that the BM as an effective approach to realize benefits can help organizations continuously identify their expected benefits and develop a plan to actualize these benefits in a well-structured process."

Response: the manuscript is revisited to remove such long sentences.

  1. The same example also illustrates a different problem within the paper, related to the non-proper use of abbreviations.

Response: We tried to solve such issue in many places

  1. I would suggest authors reproduce figure 2-1;

Response: the table is now reproduced by reformatting, and in the left side we put the BM processes and activities, and put the activities in line with the existing practices in italic. In the right side, we put the existing practices

  1. some excerpts might be revised and simplified (too long sentences), maybe splitting into two parts, improving their readability, for instance, the following: "Numerous studies confirm that while the ERP systems literature is rich in different perspectives, and while there has been considerable progress in understanding different aspects of ERP implementation, realizing business benefits from these systems at the post-implementation is problematic and puzzles several businesses, which has motivated the call for further research to address this problem [1-3,10,11,15,23]." 

Response: we tried to solve issues like mentioned above.

Round 2

Reviewer 2 Report

Dear authors,

Thanks for the suggestions made. I would only suggest improving Figure 3.2 proposed, to provide a better link between the research analysis and research outcomes in a way that readers may understand how the research questions specifically addressed could be answered through this paper.  Using codes within the figure and the methods section could also improve the link to the methodology, helping users to understand what was made and the analysis performed. 

Author Response

I would like to highly thank the reviewer for the great comments that every time strengthen the paper and provide more clarity.

As a response for the reviewer comments about the analysis, a new section is added "3.2 Data analysis" that shows the analysis procedure. Furthermore, in Table4-1 each theoretical construct (BM processes and activities) are given codes.

Just a suggestion from the authors to reduce the length of the title, by keeping the first part, and removing after ":", if that suits the reviewers

Many thanks for the consistent support